# Convolutional Neural Networks Cascade for Automatic Pupil and Iris Detection in Ocular Proton Therapy [note 1]

**DOI:** 10.3390/s21134400

**Published:** 2021-06-27

**Authors:** Luca Antonioli, Andrea Pella, Rosalinda Ricotti, Matteo Rossi, Maria Rosaria Fiore, Gabriele Belotti, Giuseppe Magro, Chiara Paganelli, Ester Orlandi, Mario Ciocca, Guido Baroni

**Affiliations:** 1Bioengineering Unit, Clinical Department, National Center for Oncological Hadrontherapy (CNAO), 27100 Pavia, Italy; andrea.pella@cnao.it (A.P.); rosalinda.ricotti@cnao.it (R.R.); guido.baroni@polimi.it (G.B.); 2Department of Electronics, Information and Bioengineering, Politecnico di Milano University, 20133 Milan, Italy; matteo2.rossi@polimi.it (M.R.); gabriele.belotti@polimi.it (G.B.); chiara.paganelli@polimi.it (C.P.); 3Radiotherapy Unit, Clinical Department, National Center for Oncological Hadrontherapy (CNAO), 27100 Pavia, Italy; mariarosaria.fiore@cnao.it (M.R.F.); ester.orlandi@cnao.it (E.O.); 4Medical Physics Unit, Clinical Department, National Center for Oncological Hadrontherapy (CNAO), 27100 Pavia, Italy; giuseppe.magro@cnao.it (G.M.); mario.ciocca@cnao.it (M.C.)

**Keywords:** ocular proton therapy, convolutional neural networks, eye tracking, pupil segmentation, iris segmentation

## Abstract

Eye tracking techniques based on deep learning are rapidly spreading in a wide variety of application fields. With this study, we want to exploit the potentiality of eye tracking techniques in ocular proton therapy (OPT) applications. We implemented a fully automatic approach based on two-stage convolutional neural networks (CNNs): the first stage roughly identifies the eye position and the second one performs a fine iris and pupil detection. We selected 707 video frames recorded during clinical operations during OPT treatments performed at our institute. 650 frames were used for training and 57 for a blind test. The estimations of iris and pupil were evaluated against the manual labelled contours delineated by a clinical operator. For iris and pupil predictions, Dice coefficient (median = 0.94 and 0.97), Szymkiewicz–Simpson coefficient (median = 0.97 and 0.98), Intersection over Union coefficient (median = 0.88 and 0.94) and Hausdorff distance (median = 11.6 and 5.0 (pixels)) were quantified. Iris and pupil regions were found to be comparable to the manually labelled ground truths. Our proposed framework could provide an automatic approach to quantitatively evaluating pupil and iris misalignments, and it could be used as an additional support tool for clinical activity, without impacting in any way with the consolidated routine.

## 1. Introduction

Radiation therapy with accelerated proton beams represents a possible solution for the treatment of ocular melanoma, and it is often considered as an alternative to surgical enucleation, photon radiosurgery or brachytherapy [1,2,3].

Ocular proton therapy (OPT) foresees a peculiar clinical workflow and requires the active participation of the patient. Prior to treatment planning, an adequate number of tantalum clips are sutured by the ophthalmologist on the sclera [4]. These radio-opaque references are used both during the planning phase (to delineate the target base contour and estimate the optimal gaze angles), as well as the treatment delivery phase, for set-up verification purposes. The gaze fixation angles are defined to optimize the dose distribution to the target while sparing critical organs and healthy tissues as much as possible. The accuracy and repeatability of setup are guaranteed during irradiation, relying on different solutions; patients’ head is firmly immobilized inside a thermoplastic mask and a fixation light is properly placed in front of the patient. By looking at this fixation light, the displacement of the eye bulb reproduces the desired gaze angles.

Before and during the irradiation, radiographic imaging is used to verify the proper alignment of the eye, by comparing the current tantalum clips configuration with the planning reference. At the same time, gaze fixation is qualitatively monitored by dedicated cameras focused on the involved eye. A clinical operator manually delineates the contours of the pupil, and eventually other structures of interest, on the images as displayed on screen. The outlined ocular features are used during real-time images acquisition as a qualitative reference to evaluate eye misalignments or shifts during irradiation. In the event of out-of-threshold misalignments, beam delivery is manually interrupted by the radiation technologist [5]. The manual definition of the reference contours, the qualitative recording surveillance, and consequently the manual gating-off, represent critical aspects for the treatment outcome. Manual beam interruption is operator-dependent; since the real-time misalignment is qualitatively evaluated, although supported by x-ray verification images acquired during the irradiation itself, the gate-off is performed with a short but inevitable delay (estimated around one second).

The National Centre of Oncological Hadrontherapy (Centro Nazionale di Adroterapia Oncologica, CNAO, Pavia, Italy) has introduced a protocol for OPT in August 2016. An accurate description of the clinical OPT treatment protocol, along with the hardware and software systems installed at CNAO, can be found in [6,7]. In particular, the CNAO OPT framework employs a custom eye-tracking system (ETS), which provides a visual reference to the patient. This system aims at maintaining gaze stability along the planned direction, and it incorporates two stereo-cameras for real-time motion monitoring.

Recent studies have investigated the introduction of new procedures in the OPT workflow for the automatic extraction of ocular features to enable quantitative monitoring of the gaze stability during treatments [8,9,10,11]. Furthermore, enormous progress in eye-tracking was made both in terms of features detection and motion analysis. The analysis of the images acquired by means of eye tracking devices provides eye-gaze estimation that can be used in several fields such as medicine, marketing, engineering and gaming [12]. For close-up eye images analysis, convolutional neural networks (CNNs) have been demonstrated to be the most suitable, as they ensure high accuracy and robustness [13].

The majority of deep-learning based eye trackers aim at performing well in different setup and environment circumstances: CNNs are trained with datasets as large and heterogeneous as possible in order to obtain a predictor as flexible as possible and able to detect features in most situations, especially in unconstrained lighting conditions or under low resolution constraints [14,15]. Furthermore, recent publications highlighted the potentiality of CNNs for eye feature extraction [16,17,18,19,20,21,22,23,24].

As already mentioned, several CNN architectures have been implemented and proposed in the field of medical image segmentation [25], but the one that has shown greater adaptability and better generalization capacity is the U-Net architecture [26]; a visual representation is depicted in Figure 1. U-Net is characterized by an encoder–decoder architecture, and consists of convolutional layers with a series of down-sampling followed by progressive up-sampling blocks with skip connections between the two paths. The left side, the encoder path, consists of 3 × 3 convolutional layers followed by batch normalization and Leaky Relu activation functions. A 2D max-pooling layer follows these operations. Each block in the encoder path doubles the feature maps and halves the image size. The right side, the decoder path, performs the same operations as the encoder one, but the max-pooling is replaced with a 2D up-sampling operation. Each block in the decoder path halves the feature maps and doubles the image size. Furthermore, the decoder path also performs a concatenation operation between the up-sampled feature maps of the lower blocks and the output of the encoder path at the same level. This concatenation guarantees that the information is properly propagated between the two paths, allowing the recovery of spatial information, which has been lost during the max-pooling operations. Finally, the output layer is a single filter 2D convolutional layer with a 1 × 1 kernel. This layer also employs a sigmoid activation function, which ensures that the values of the probabilistic output map are comprised between 0–1.

We recently discussed a preliminary study aimed at evaluating the feasibility of pupil automatic segmentation procedures based on feature extraction [27]. Here, we propose a significant step forward to a more sophisticated approach, taking advantage of CNNs for iris and pupil detection on the eye surface images extracted from surveillance videos recorded by ETS device used at CNAO during OPT clinical workflow. The new method is based on a dual stage CNNs framework that locates the pupil and iris with three cascaded U-Nets from coarse to fine localization. In the first stage, a region of interest (ROI) is automatically extracted and served as an input for the U-Nets, and a second pipeline stage aimed at conclusive pupil and iris detection.

## 2. Materials and Methods

### 2.1. Patient Dataset

Clinical data of 140 ocular melanoma patients treated at CNAO between January 2018 and December 2020 were retrospectively collected. The study was performed within the Local Ethics Committee notification (notification n° 37143/2021). The patients gave their written informed consent for ocular proton treatment, and use of their anonymized data for educational and research purposes. 

Patient compliance to treatment participation was considered clinically acceptable. The mean (std deviation) age of the patient cohort was 61 (13) years. As highlighted in Table 1, in 110 patients the eye surface surveillance videos recorded the diseased eye looking at the fixation light (left eye: 61 patients; right eye: 49 patients). Whereas, 30 patients showed visual impairment of the diseased eye, and the contralateral eye (left eye: 14 patients, right eye: 16 patients) was used for gaze stabilization.

ETS infrared videos were recorded in RGB24 AVI format, with a resolution of 512 × 640 pixels and a frame rate ranging between 6–12 Hz. Images were stored in real-time in a dedicated work-station at the time of treatment, and made available for off-line analysis.

For the proposed method, the training dataset included 650 randomly selected video frames recorded by the ETS during different OPT clinical procedures (treatment preparation and delivery) for 120 patients. An independent dataset of 57 video frames of 20 patients treated with OPT at CNAO was used to evaluate the performance of the method. Overall, we considered data of 140 patients, and we extracted a total of 707 images from the surveillance videos (training set = 650 images; test set = 57 images).

The reference pupil and iris segmentation were manually contoured on each video frame by a clinical operator, and were used as ground truth during training and testing. In addition, a region of interest (ROI) of 350 × 350 pixels that embodied the pupil and iris was manually selected. 

### 2.2. Framework Design and Training Details

The proposed framework used the same U-Net structure for three different tasks, which operated in cascade with respect to each other. The first one aimed at roughly identifying the eye within the original image and at extracting an ROI around it. The U-Net training for this first task, namely the ROI U-Net, was performed using the original frames without any pre-processing operations and extracted directly from the ETS recordings (Figure 2a). The corresponding binary masks are depicted in Figure 2c and are composed by merging iris and pupil structures (Figure 2b). As will be explained in the U-Nets cascade section, the second and third tasks consisted respectively of iris and pupil detection within an ROI extracted from the original image. These networks are called Pupil U-Net and Iris U-Net, respectively. The ROI is a smaller portion of the image that should include both the structures and excludes background features. In order to maximize the iris and pupil prediction capabilities, the training process was performed by using the 350 × 350 pixels ROI manually extracted from the video frames. Accordingly, the same ROI was identified within the masks. This operation allowed the definition of the ground truth for the two training processes. Figure 2, in panel (d) and (e), depicts the ROI-masks used respectively for the iris and pupil training procedures. The dimensions of the binary masks were 350 × 350 pixels.

### 2.3. U-Net Details

For the scope of our work we adapted the original U-Net proposed by Ronneberger et al. [26] (Figure 1). We investigated the benefits of reducing the feature map number without changing the blocks number. The original U-Net [26] aimed at identifying cells with complex geometry, and adopted 64 filters in the first layer and 1024 in the last one. The choice of filter number depends on the complexity of the objects which have to be recognized by the model. Since the iris and pupil have basic shapes (mostly circular/elliptical), for each U-Net we tested 3 configurations of feature maps: 16, 32, and 64. The best one has been identified through the analysis of accuracy and loss curves (Figure 3). Each network has been trained for 60 epochs and the cross-entropy function has been selected as loss function. The 3 networks (Pupil, Iris, and ROI U-Net) behaved in a similar way. In particular, Figure 3 depicts accuracy and loss curves for the Pupil U-Net: after 60 epochs the 3 different configurations converged with different speeds to the same value, both in terms of accuracy and loss. We preferred the lighter configuration, promoting speed and reducing the inference time.

The total number of trainable parameters is 1,947,153. Input image size has been fixed to 256 × 256 pixels. 

Once the architecture had been defined, the hyperparameter search had been refined through an iterative manual approach. We decided to adopt a batch size equal to 4 to speed up the training process, as smaller batches make it easier to fit one batch worth of training data in memory. The weight optimization was done through the Adaptive moment estimation (Adam) with a constant learning rate of $10^{-4}$ and zero decay. The binary cross-entropy function was chosen as a loss function. The best weights were selected by evaluating the performance on the training set in terms of cross-entropy. 

In each training routine, all of the hyperparameters were maintained constant.

To face a limited dataset and to make the model more robust, data augmentation techniques were used on the training set. The following operations were randomly applied to each pair of image-masks during the training process:Reflection about the vertical and horizontal axisRandom clockwise rotation from 0° to 360° degreesImage translation in both axis of 0–70 pixelsImage zoom by a factor ±0.2

### 2.4. U-Nets Cascade

As stated in the introduction, we attempted to derive a robust detection of the pupil and iris by concatenating cascades of U-Nets. The overall workflow is shown in Figure 4. Overall, the entire method can be divided into two parts: the first block consisted in the localization of the eye inside the original image and in the extraction of an ROI around the eye (ROI U-Net). In the second one, the pupil and iris were detected inside the region of interest, and the predictions were mounted back on the original image (Pupil U-Net and Iris U-Net).

The input image, with size 512 × 640 pixels, was rescaled to 256 × 256 pixels and gave as input to the ROI U-Net. The output probabilistic map had a threshold at a probability of *p* > 0.2, denoised through morphological closing, and the largest connected component was extracted to reject small false positive regions. The binarized mask was then rescaled to the original image size.

Once the eye was located in the image, it was possible to identify a window around it, which was applied to the original image for ROI extraction. The centroid of the ROI was computed, and the binary prediction was cropped by a 350 × 350 pixels squared window centered on the ROI centroids coordinates (Figure 5 red dots). Its size was selected to be large enough to guarantee the inclusion of all of the ocular features within the window. In case the eye was located at the edges of the image, the cropping ROI exited the image as depicted in Figure 5b. In this case, the window was translated along the axis until the window was completely inside the image (blue square).

The ROI image size was then reshaped to 256 × 256 pixels and given as input, firstly to the iris U-Net, and secondly to the pupil U-Net. Both the probabilistic maps had thresholds at a probability of *p* > 0.5 and *p* > 0.15, respectively. Morphological closing and largest connected component detections were computed to exclude small false positive regions eventually predicted by the network. A reshape operation was performed to recover the ROI original size (350 × 350 pixels). Finally, the two prediction masks were merged together and brought back to the correct position inside the original image.

### 2.5. Evaluation Metrics

The training was carried out in a Cuda-enabled environment, equipped with a 4-core CPU, 25 GB RAM and a NVIDIA TESLA T4 GPU card. The neural networks implementation and their training routines were computed with Python, leveraging the Keras libraries and the Tensorflow framework [28]. Model evaluations were computed on the test dataset. It consisted of blind data (57 frames) not used for network training. 

To assess the accuracy of the ROI U-Net, we evaluated the Euclidean distance, measured in pixels, between the predicted eye center and the ground truth label. 

Iris U-Net and pupil U-Net predictions were compared with the corresponding masks, and evaluated with some commonly accepted measures of quality for segmentation tasks [29]: DICE coefficient, Szymkiewicz–Simpson coefficient, Intersection over Union (IoU) and Hausdorff Distance. 

The DICE coefficient computes the overlap between predicted and manually labelled areas, and assuming that A is the segmentation performed by the model and G is the ground truth is defined as:$$\mathrm{DICE}=\frac{2\left|A\;\cap \;G\right|}{\left|A\right|+\left|G\right|}$$

The Szymkiewicz–Simpson coefficient expresses the ratio between the size of the intersection and the smaller size of the two areas:$$\mathrm{Szymkiewicz}\text{–}\mathrm{Simpson}=\frac{\left|A\;\cap \;G\right|}{\min\left(\left|A\right|,\left|G\right|\right)}$$

The IoU expresses the size of the intersection divided by the size of the union of the sample sets:$$\mathrm{IoU}=\frac{\left|A\;\cap \;G\right|}{\left|A\;\cup \;G\right|}$$

All of these similarity coefficients range between 0 and 1. The Hausdorff distance, expressed in pixels, measures the maximum contour distance between prediction and manual labelling.

Moreover, the entire framework was also evaluated considering the total execution time to explore the feasibility of its introduction in the clinical workflow.

## 3. Results

### 3.1. ROI U-Net

This network received the original image as input and predicted the eye position, resulting in the eye center coordinates. To evaluate its performance, we computed the Euclidean distance between the predicted eye center and the ground truth label. For the 57 frames included in the test set, median (IQR) Euclidean distance, was 3.05 (4.14) pixels. Figure 6 illustrates some exemplary cases.

### 3.2. Iris U-Net and Pupil U-Net

Both networks were evaluated with the previously described metrics. Their input was an ROI extracted from the original image, and their output was a ROI probabilistic map with pupil and iris predictions. We decided to apply the metrics not on the ROI predictions but over the entire image: before the evaluation, the ROI probabilistic maps were transferred back to the original image location.

Figure 7 depicts on the first line (a), the eye surface images and, on the second one (b), the final output of the iris and pupil U-Nets super-imposed on the original images.

The evaluation metrics distributions are summarized as boxplots in Figure 8. Table 2 reports their values in terms of median and interquartile ranges.

### 3.3. Inference Time

We tested our entire framework on different machine setups with and without GPU to evaluate the total inference time. We computed the total inference time of this framework, starting from the original image and applying the prediction masks onto it. Then, we divided the entire task into five main subtasks, and we computed the execution time of each one to investigate which operations were time-consuming. The ROI extraction process was divided into two subtasks: the first one consisted of the ROI prediction performed by the ROI U-Net, while the second one included the post-processing operations performed on the probabilistic map, and consequently the extraction of a 350 × 350 pixel window from the original image. Pupil and iris detection processes were considered two further distinct subtasks. The last one was the set of operations performed on the iris and pupil U-Net predictions (binarizations, morphological closing and reshape to original dimensions).

As reported in Table 3, the machines equipped with GPU performed better. The total mean execution time to process one image without GPU was 366 ms. Using a GPU with compute capability of 3.7, the execution time dropped to 339 ms, while a 7.5 compute capability GPU lowered the time further to 249 ms.

## 4. Discussion

The presented study proposed and evaluated a new approach for the automatic pupil and iris detection on the eye surface images extracted from surveillance videos recorded by the ETS device used at CNAO during OPT clinical workflow. The presented method was based on a dual stage convolutional neural network pipeline that located the pupil and iris with two cascaded U-Nets from coarse (ROI U-Net) to fine localization (Pupil U-Net and Iris U-Net). In its first stage, the pipeline performed rough eye localization within the video frame using the presented U-Net architecture for the ROI identification (ROI U-Net). The extracted ROI embodied the pupil and iris, and it was used as an input for the U-Nets in the second pipeline stage (pupil U-Net and iris U-Net) aimed to provide conclusive pupil and iris detection. 

The ROI identification stage was devoted to extract a portion of image in which the relevant ocular features (pupil and iris) should be likely included thus discarding eventual background elements such as eyelid, eyelashes and retractors. Moreover, this cropping operation helped to reduce eventual illuminating inhomogeneities, mitigating the impact of image noise and permitted a larger pupil and iris resolution. In fact, the last two U-Nets (pupil U-Net and iris U-Net) operated on a larger pupil and iris resolution providing an increased feature extraction accuracy. The detection of the eye center was improved by a 7% factor between the first and the second stage of the proposed method. Several CNN approaches for object classification within an image increased their performance by restricting the recognition problem to a given region of the image [19,30,31]. In our proposed method, similarly to the one proposed by Fuhl et al. [19], the ROI extraction does not require manual interaction, and it automatically provides the input image for the subsequent U-Nets. 

We decided not to develop a multiclass U-Net but we opted for single class networks for two main reasons: (I) we believe that our dataset was larger enough for training single class networks but not for multiclass. (II) The identification and discrimination of features requires heavier and deeper networks. As we wanted a framework that could potentially have capabilities for real-time analysis, we decided to use two faster and lighter structures rather than one heavier. Moreover, the design of the proposed method is flexible enough for permitting eventual future integrations of additional U-Net cascades aimed at other supplementary ocular feature extractions.

In 51 out 57 testing video frames (90% of the testing dataset) Dice, Szymkiewicz–Simpson and IoU coefficients were greater than 0.80 for both pupil and iris automatic detection. The selected similarity metrics suggested that the detected pupil and iris areas were highly similar to the manually labelled ground truths. Pupil detection performed better than iris detection, reporting greater median similarity coefficients. In addition, the interquartile ranges of the similarity coefficient distributions were found to be smaller for the automatic pupil detection rather than for the automatic iris detection: in our opinion this confirmed the robustness of pupil extraction against different testing video frames. Moreover, the automatic estimation of iris areas were slightly overestimated with respect to the manual contours, as suggested by IoU coefficients. Besides, the Hausdorff distance of iris contours was greater than the pupil, and showed greater variability. Although, there were a few failure cases in the test set as reported in Figure 9. The main causes may depend on the lack of sufficient contrast between the sclera-iris and pupil-iris structures: the ETS position into the space affects the scene illumination, and thus how much two structures are discernible. Furthermore, we have noticed that the network performance worsened when the images were blurred or when the eye was positioned at the image borders. Not surprisingly, also in clinical practice, the more the video images are blurred, the more the inter-operator variability increased [27]. 

Image quality is strongly affected by several factors which are introduced by the OPT clinical workflow:Patient is immobilized on a treatment chair by using a personal thermoplastic mask;Eye retractors are placed to prevent eyelid movements and to maintain the eye wide open;Room illumination may be not homogenously diffused (this is a patient specific variable, since some of them are facilitated by low light conditions. ETS features two IR LEDs, resulting in insensitivity in those cases)Camera FOV is intrinsically not constant, depending by the desired gaze direction and the displacement of ETS in refer to the patient.

As our CNN method has been accurately trained on a dataset that includes all the above-mentioned peculiarities, we decided not to consider a comparison with other state-of-the-art algorithms. Although they have been trained to be as flexible as possible and to have a high generalization ability, our dataset possesses unique characteristics, and we believe that a benchmark with our method would be unfair.

The OPT treatments at CNAO strongly rely on the expertise of the clinical operator who manually delineates the contours of the main structures on images recorded by means of the ETS and, depending on the qualitative evaluation of misalignments and supported by x-ray verification images acquired during the irradiation, manually interrupts beam delivery. 

Manual segmentation procedures are operator-dependent and time-demanding. As reported in Table 2, our method could potentially perform image segmentation in less than 400 ms using PCs without GPU, and less than 250 ms in PCs equipped with GPU. During treatment, we can assume that the ROI selection on screen is well maintained, so that the eye position inside the 512 × 640 pixel images should not change significantly. This suggests that the first step of the U-Net cascade could be executed only during the first frames. Once the ROI position is defined, the reduction of total inference time can be achieved by running only the iris and pupil detection pipeline.

## Figures and Tables

**Figure 1 sensors-21-04400-f001:**
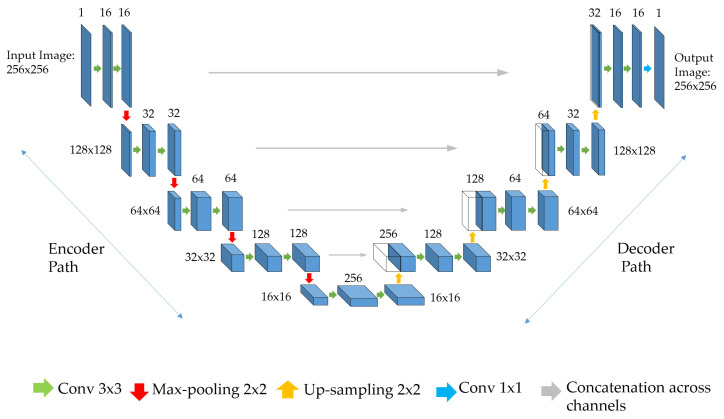
U-Net architecture. Each map’s dimensions are specified on the left side for the encoder path and on the right side for the decoder path. The number of channels is indicated above each blue box. Both the input image (leftmost map) and the output prediction (rightmost map) are 256 × 256 pixels. As specified in the legend, green and blue arrows represent convolutions, red and yellow represent max-pooling and up-sampling operations, while the grey ones represent concatenation operations.

**Figure 2 sensors-21-04400-f002:**
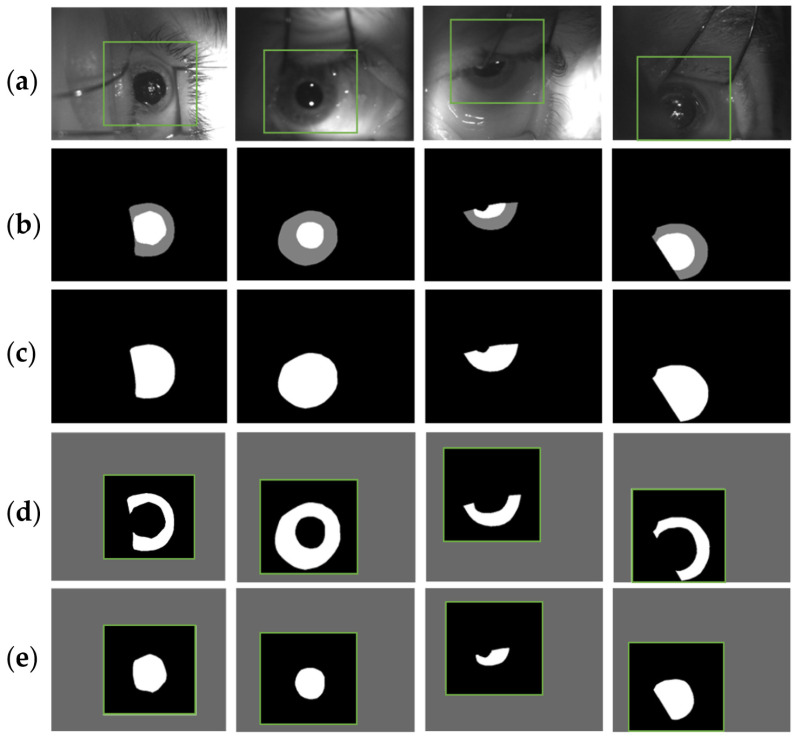
(**a**) Original images used for U-Net training and testing. They were extracted directly from ETS recordings. The green squares identify the ROI areas that have been manually selected by the clinical operator. They were used during the U-Net training and testing for iris and pupil detection. (**b**) Manual contours of iris (grey) and pupil (white). (**c**) Ground truth images in which iris and pupil structures are merged together and used for training and testing the ROI U-Net. (**d**,**e**) iris and pupil binary ROIs extracted from the original masks. They were used for training and testing the iris and pupil U-Nets.

**Figure 3 sensors-21-04400-f003:**
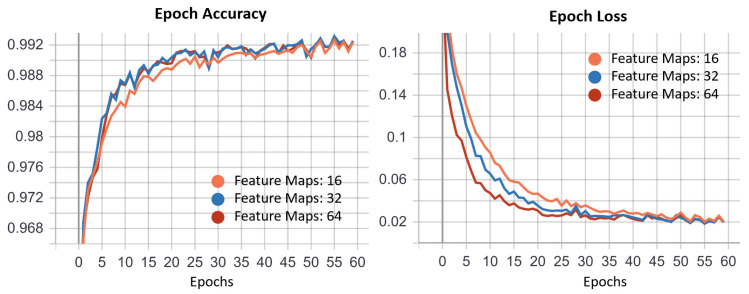
We trained our model for 60 epochs and we tested the same architecture varying the number of feature maps in the first layer for 16, 32, and 64. The loss function used is the cross-entropy function.

**Figure 4 sensors-21-04400-f004:**
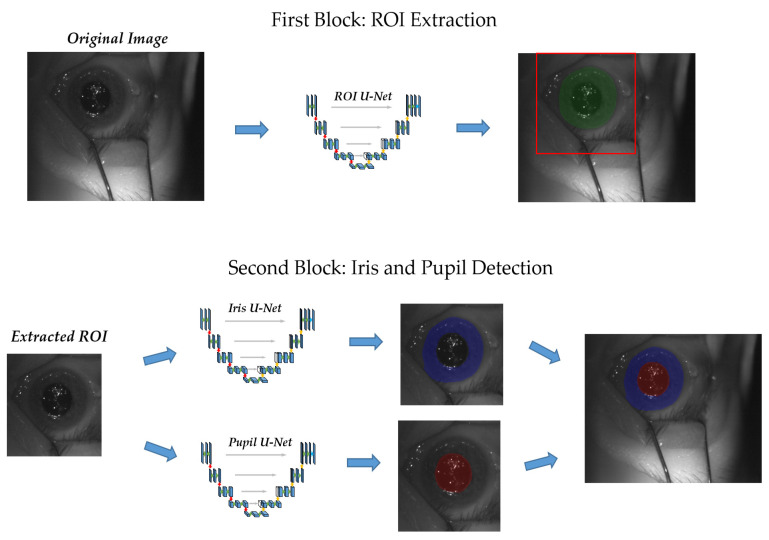
The process can be divided into two blocks: the first one, after having localized the eye inside the image (green prediction), identifies a 350 × 350 pixel ROI (red square), and the second one detects the iris and pupil area (blue and red predictions, respectively).

**Figure 5 sensors-21-04400-f005:**
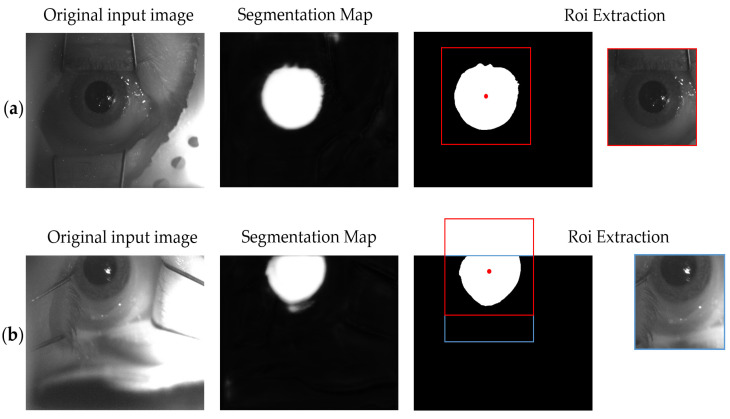
The original image is fed to the U-Net, which generates a probabilistic map. Once binarized and applied the closing operations, the centroid is calculated and a square window (350 × 350) centered on it is defined and applied to the original image. (**a**) No translation operations are required because the window is within the image. (**b**) It is necessary to translate the window downward.

**Figure 6 sensors-21-04400-f006:**
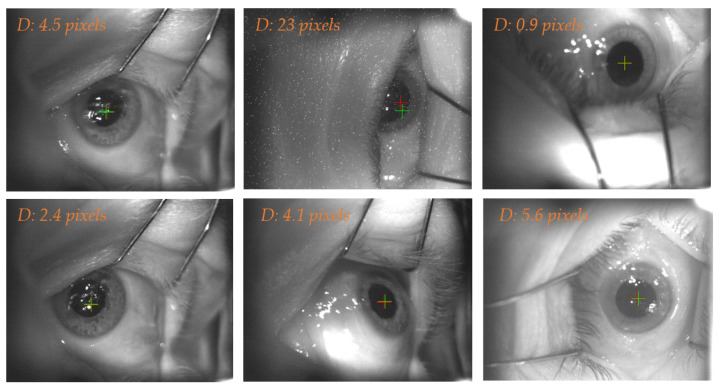
The crosses identify the eye center computed from the manually labelled eye (red) and the ROI U-Net predictions (green).

**Figure 7 sensors-21-04400-f007:**
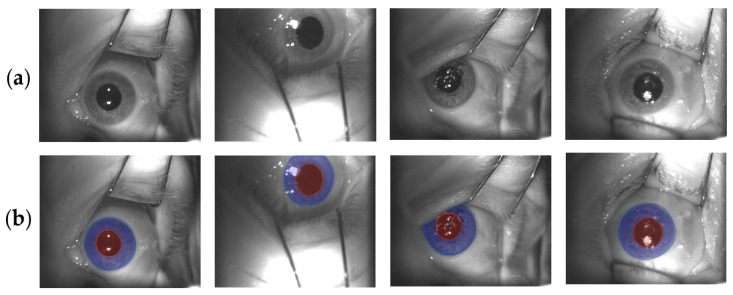
The original images (**a**) and final U-Net predictions (**b**). The predicted iris and pupil areas are depicted in blue and red, respectively. The red and blue continuous lines are the ground truth contours of pupil and iris, respectively.

**Figure 8 sensors-21-04400-f008:**
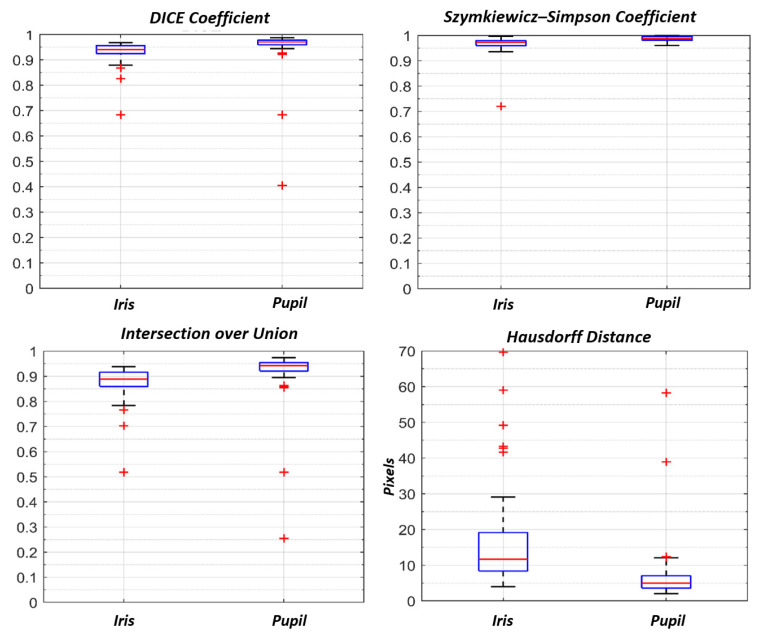
Boxplots of Dice, Szymkiewicz–Simpson, IoU and Hausdorff Distance (expressed in pixels). Iris and pupil predictions were performed with the iris U-Net and the pupil U-Net, respectively, on an ROI image. The metrics were evaluated once the 350 × 350 pixels probabilistic maps were restored on the original image size (512 × 640).

**Figure 9 sensors-21-04400-f009:**
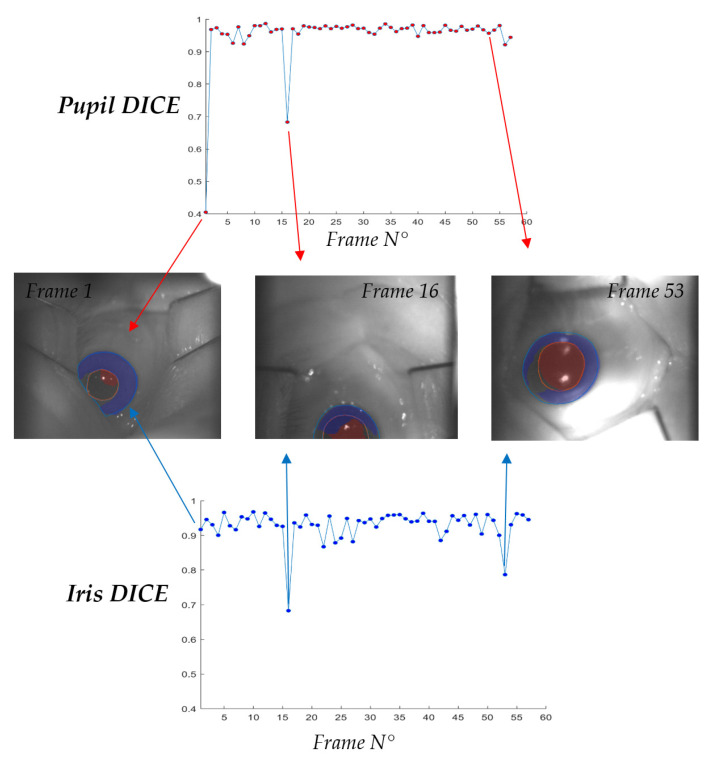
Graphical representation of the Dice coefficients coming from the comparison between the predicted and manually contoured structures (iris and pupil) for the 57th frames present in the test set. Furthermore, the figure depicts examples of failed pupil segmentation (frame 1 and 16) and failed iris segmentation (frame 16 and 53). In frame n° 1 iris prediction worked correctly (iris Dice = 0.92) while pupil prediction failed (pupil Dice = 0.41). On the other side, frame 53 presents an iris Dice = 0.76 while pupil Dice = 0.94. The worst case is the frame 16 in which both the Pupil and Iris U-Net underperformed (Pupil Dice = 0.68 and Iris Dice = 0.68).

**Table 1 sensors-21-04400-t001:** Patient cohort description.

Fixation Eye	N. of Patients	Left Eye	Right Eye
Diseased	110	61	49
Contralateral	30	14	16

**Table 2 sensors-21-04400-t002:** Difference between predicted and manually labelled iris and pupil regions, measured in terms of median values (75–25 percentile) of DICE coefficient, Szymkiewicz–Simpson coefficient, IoU coefficient and Hausdorff Distance.

	DICE Coefficient	Szymkiewicz–Simpson Coefficient	IoU Area	Hausdorff Distance (Pixel)
Iris	0.94 (0.96–0.92)	0.97(0.98–0.96)	0.88 (0.92–0.86)	11.7 (19.1–8.4)
Pupil	0.97 (0.98–0.96)	0.99 (0.99–0.98)	0.94 (0.96–0.92)	5.0 (7.0–3.6)

**Table 3 sensors-21-04400-t003:** Execution time of the five subtasks identified during the entire workflow, along with the total inference time. Values are computed over the entire test set and averaged over the total number of images. They are expressed in terms of median values (75–25 percentile). Best performances are obtained by Tesla T4 (compute capability 7.5).

	Tesla K80 CC: 3.7	Tesla T4 CC: 7.5	No GPU
ROI Prediction	105 (110–103) ms	105 (110–103) ms	121 (124–117) ms
ROI Post-Processing	44 (45–43) ms	37 (38–37) ms	29 (30–29) ms
IRIS Prediction	67 (69–61) ms	45 (45–43) ms	93 (95–90) ms
Pupil Prediction	73 (74–71) ms	47 (48–45) ms	93 (95–90) ms
Final Post-processing	50 (51–49) ms	43 (45–42) ms	30 (31–29) ms
Total Inference Time	338 (340–335) ms	249 (251–245) ms	366 (371–359) ms

## Data Availability

Not Applicable.

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
