# Peer review of "Convolutional Neural Networks Cascade for Automatic Pupil and Iris Detection in Ocular Proton Therapy†"

_sensors, 2021, doi:10.3390/s21134400_

Round 1

Reviewer 1 Report

Dear authors, 

I have finished the review of your paper. Although there are interesting results and your paper is original, I have the following concerns.

  1. Include info about additional parameters like size of batch in the experimental setup.
  2. You must make a comparison between your model and at least one featured model of the state of art, since you are testing with your own datase, it would be very useful to see which method performs better.
  3. Also consider the execution time for the comparison.
  4. One minor issue is: add the mathematical expression for the metrics used in the results section.

I hope you find these recommendations useful for your paper.  

Reviewer 2 Report

This work proposes a new approach for eye tracking in ocular proton therapy (OPT). It is based on two stages of convolutional neural networks (CNNs): the first stage identifies the eye position and the second stage performs a fine iris and pupil detection.

I think that the quality of this work is good, but there are several points that can be improved:

  1. In the Introduction section, the U-Net discussion should reference to Fig. 1 for a better understanding of the architecture.
  2. In Section 2.2, the parameters of the U-Net architecture are described. Why was this specific architecture selected as the one to perform all the experiments? Did you do any type of experiments regarding the best selection of hyper-parameters of the U-Net?
  3. In Section 2.4, I understand that the image has to be reshaped for being processed by the U-Net. However, it seems that there are two stages of reshaping. I think that one stage of reshaping could be avoided if the architecture of the U-Net was designed to process 350x350 images.
  4. Regarding the two previous points, my general understanding of the proposed method is that the U-Net was taking from the original paper, but it was not fully adapted to the needs of this work. For instance, it could be adapted to process 350x350 images directly. Also, there could be some experiments showing which is the best architecture of the U-Net for each of the two stages of the detections.
  5. In the experiments section, the performance of the proposed method should be compared with other state-of-the-art systems, maybe based on other types of neural network architectures. This experiment would be very interesting in order to understand the advantage of using the proposed method compared to other existing methods.

Reviewer 3 Report

The manuscript presents an eye-tracking algorithm based on a deep learning architecture that works on OPT images. The results seem promising and accurate, however, in my opinion, the following ideas may improve the quality of the paper:

A) Open Science and datasets.

Providing the datasets and liberating them to other researchers may allow other parties to build and test better applications. The authors should indicate whether this data is available and where.

B) Detection

Is the pupil and iris detected in every frame of the dataset? Figure 7 shows the error, but it may be expected to loose track of the pupil in some sequences. If this is the case, precision and recall should be computed. If possible, a precision-recall figure should be created based on some parameters of the algorithm.

C) Description of the dataset

The description of the dataset should be more detailed. In Section 2.1 the total number of frames used for training and the total number of frames used for testing should be mentioned.

D) Section 2.2

The design of the architecture should be justified. How did the authors find that this architecture is optimal in any way?

Minor:

Line 101 (std) --> std deviation

Round 2

Reviewer 1 Report

Dear authors,

I have reviewed your revised version. I have no more recommendations or concerns about this work.